# Human Papillomavirus Infection of the Oropharyngeal and Laryngeal Squamous Papilloma: Disparities in Prevalence and Characteristics

**DOI:** 10.3390/diagnostics14111163

**Published:** 2024-05-31

**Authors:** Jihye Kwak, Dongbin Ahn, Mee-seon Kim

**Affiliations:** 1Department of Otolaryngology-Head and Neck Surgery, School of Medicine, Kyungpook National University, Daegu 41944, Republic of Korea; laugh112@naver.com; 2Department of Pathology, School of Dentistry, Kyungpook National University, Daegu 41944, Republic of Korea; kimm23@naver.com

**Keywords:** squamous papilloma, human papillomavirus, oropharynx, larynx, prevalence, genotype

## Abstract

Human papillomavirus (HPV) infection has emerged as an etiologic factor of squamous papilloma (SP). The oropharynx and larynx are common sites of SP, but studies on the prevalence of HPV infection in these sites are lacking. This study aimed to evaluate and compare the prevalence and characteristics of HPV infection in oropharyngeal SP (OPSP) and laryngeal SP (LSP). HPV detection and genotyping data of patients with pathologically confirmed OPSP and LSP were retrospectively analyzed. A total of 119 patients were enrolled, consisting of 93 patients with OPSP and 26 patients with LSP. Of those patients, 13 patients with OPSP and 14 patients with LSP were positive for HPV infection, accounting for a prevalence of 14.0% and 53.8%, respectively (*p* < 0.001). The most prevalent genotype was HPV16 in OPSP and HPV6 in LSP. Over two-thirds (69.2%) of HPV(+)-OPSP infections were high-risk types compared with 14.3% of HPV(+)-LSP infections (*p* = 0.004). The prevalence of HPV infection in patients with OPSP and LSP demonstrated no differences in terms of age, sex, and smoking status. These results could provide a better understanding of HPV infection in OPSP and LSP and serve as a background for the epidemiology of HPV-related tumorigenesis of the oropharynx and larynx.

## 1. Introduction

Head and neck squamous papilloma (SP) is a commonly encountered benign epithelial tumor characterized by fronds or papillary projections that originate from the central fibrovascular stroma [1,2,3]. Although mechanical and chemical irritations have traditionally been associated with SP development, the exact etiology remains unknown [2]. Since the 1980s, human papillomavirus (HPV) has emerged as an etiologic factor based on epidemiological and experimental data [1,4,5].

The oropharynx and larynx, prominent sites of head and neck SP, are integral components of the head and neck region that harbor the potential for HPV infection [6,7]. The characteristics of HPV infection in these sites have been evaluated with distinct clinical associations [8,9,10,11]. The causative role in the etiopathogenesis of oropharyngeal squamous cell carcinoma (OPSCC) in the oropharynx is being highlighted [8,9], and the increasing incidence of HPV-related OPSCC warrants further understanding of epidemiological data on oropharyngeal HPV infection [8,9,12]. Recent investigations into oropharyngeal SP (OPSP) have revealed a 14.5% overall prevalence of HPV infection, with high-risk (HR)-HPV accounting for 75% of all HPV-positive cases and HPV16 emerging as the predominant genotype [13]. The larynx is another HPV-tropic site mainly associated with laryngeal SP (LSP). The prevalence of HPV infection reportedly ranges from 83% to 100%, featuring predominantly low-risk genotypes such as HPV6 and HPV11 [10,11,14,15]. Nevertheless, a significant research gap persists, as the majority of previous studies on LSP have focused on recurrent respiratory papillomatosis (RRP), which is a specific clinical manifestation characterized by multiple confluent LSPs, thereby leaving a lack of data for all types, including solitary LSP [16]. The histological characteristics of SP are identical regardless of anatomical sites. Still, potential discrepancies in the prevalence and characteristics of HPV infection are encountered across these two anatomical sites, which requires further investigation of HPV infection in head and neck SP.

Therefore, this study aimed to investigate the prevalence and characteristics of HPV infection in OPSP and LSP and compare them in terms of anatomic site, thereby providing valuable epidemiological data on HPV infection in SP.

## 2. Materials and Methods

### 2.1. Study Design and Patients

The Institutional Review Board of our institution approved the protocols of this cohort study. They waived the requirement for informed consent because of the retrospective study design.

This study included patients with pathologically confirmed OPSP and LSP, which were surgically removed for curative intent in our hospital. The exclusion criteria were (1) unavailable HPV status evaluation, (2) HPV vaccination history, (3) previous head and neck cancer (HNC), and (4) incomplete medical records.

### 2.2. HPV Detection and Genotyping

CLART HPV 2 (Genomica, Madrid, Spain), a commercial low-density microarray kit based on the polymerase chain reaction (PCR), was used for HPV DNA detection and genotyping. It enables the identification of 35 different HPV genotypes, 18 HR-HPVs (16, 18, 26, 31, 33, 35, 39, 45, 51, 52, 53, 56, 58, 59, 66, 68, 73, and 82), and 17 low-risk (LR) HPVs (6, 11, 40, 42, 43, 44, 54, 61, 62, 70, 71, 72, 81, 83, 84, 85, and 89).

DNA was extracted from serially sectioned tissue blocks from formalin-fixed paraffin-embedded tissue, and purified DNA was used for PCR amplification. A 450-bp genotype-specific HPV L1 fragment was amplified along with two internal controls, including a fragment of human CFTR that was used as a genomic DNA extraction control and a modified plasmid that was utilized as an amplification reaction control. The detection results were visualized on a CLART microarray, and hybridization between the amplicons and type-specific probes produced an insoluble precipitate, which was analyzed on a Clinical Array Reader (Genomica, Madrid, Spain).

### 2.3. Histopathological Examination and p16 Immunohistochemistry

The tissue samples were stained with hematoxylin and eosin for histopathological examination. A specialized head and neck pathologist performed a pathological diagnosis of OPSP and LSP. Additionally, immunohistochemistry (IHC) of p16 overexpression, which was used as a surrogate indicator for transcriptionally active HR-HPV in SP, was performed using CINtec^®^ p16 Histology (Ventana, Medical System, Tucson, AZ, USA). HPV-positive OPSCC with high p16 expression was utilized as a positive control, and tumors with ≥70% strong and homogeneous staining of the nucleus and cytoplasm compared with the positive control were defined as p16 overexpression. Other staining patterns were defined as negative. In some SPs without HPV infection, p16 IHC was also performed as a negative control to confirm the negative results of p16 IHC.

### 2.4. Assessment Parameters and Statistical Analysis

The primary study outcomes were the overall prevalence of HPV infection and detected genotypes in OPSPs and LSPs. The overall prevalence was calculated by dividing the number of PCR-positive patients by the total number.

The prevalence of HPV infection was assessed according to the patients’ clinicodemographic characteristics, such as age, sex, smoking status, tumor focality, recurrence, and anatomical subsites of the oropharynx and larynx. Additionally, the associations between the prevalence of HPV infection and clinicopathological characteristics were evaluated.

Baseline characteristics, including age, sex, smoking status, clinical presentation, tumor focality, presence of dysplasia on pathological examination, recurrence after treatment, and p16 IHC, were assessed and compared between patients with OPSP and those with LSP. Furthermore, clinicodemographic characteristics, such as age, sex, smoking status, tumor focality, recurrence, and genotype distribution (LR and HR), were compared between HPV(+)-OPSP and HPV(+)-LSP.

Statistical Package for the Social Sciences version 18.0 (SPSS Inc., Chicago, IL, USA) was used for statistical analysis. Continuous data were expressed as mean ± standard deviation or median (range), and categorical data were expressed as the number of patients (%). The independent *t*-test was used to compare continuous data, and the chi-square test and Fisher’s exact test were utilized to compare categorical data. Results with a two-sided *p*-value of <0.05 were considered statistically significant.

## 3. Results

### 3.1. Clinicodemographic Characteristics

Figure 1 presents the patient enrollment flowchart. This study initially assessed 284 patients with head and neck SP from April 2016 and June 2023 for eligibility. A total of 119 patients were enrolled and analyzed, of whom 93 (78.2%) and 26 (21.8%) had OPSP and LSP, respectively. The patients with OPSP demonstrated a mean age of 50.3 ± 16.5 years and comprised 61 (65.6%) males and 32 (34.4%) females. The patients with LSP demonstrated a mean age of 55.3 ± 15.5 years and consisted of 23 (88.5%) males and 3 (11.5%) females. The proportion of males and females demonstrated a significant difference (*p* = 0.024), whereas the mean age did not differ between OPSP and LSP (*p* = 0.171).

OPSP was found in 64 (68.8%) patients with no tumor-related symptoms, whereas 22 (23.7%), 5 (5.4%), and 2 (2.1%) patients had foreign body sensation, sore throat, and chronic cough, respectively. LSP was incidentally found in 10 (38.5%) patients during endoscopic examination, whereas 14 (53.9%), 1 (3.8%), and 1 (3.8%) patient(s) had voice change, foreign body sensation, and chronic cough, respectively. The proportion of symptomatic patients in the LSP group was significantly higher than that in the OPSP group (31.2% vs. 61.5%, *p* = 0.005). OPSP in 87 (93.5%) and 6 (6.5%) patients and LSP in 16 (61.5%) and 10 (38.5%) patients were solitary and multiple, respectively. The proportion of solitary SP was significantly higher in OPSP than in LSP (*p* < 0.01). Dysplasia on pathologic examination or postoperative recurrence was not observed in the OPSP group. However, dysplasia on pathological examination and recurrence after excision once or more times were observed in six (23.1%) and seven (26.9%) patients with LSP, respectively (Table 1).

### 3.2. Prevalence of HPV Infection and Genotype Characteristics

Among 93 patients with OPSP, 13 were positive for HPV infection, accounting for a prevalence of 14.0%. Among 26 patients with LSP, 14 were positive for HPV infection, accounting for a prevalence of 53.8%. The difference in the prevalence of HPV infection was statistically significant between OPSP and LSP (*p* < 0.001).

Overall, seven HPV genotypes were detected in OPSPs, of which three were LR-HPVs and four were HR-HPVs. HPV6, HPV11, and HPV84 were detected in one, two, and one patient(s) among the LR-HPV types, respectively. HPV16, HPV58, and coinfection with HPV39 and HPV66 were detected in seven, one, and one patient(s) among the HR-HPV types, respectively. Concerning LSP, five HPV genotypes were found in LSPs, of which three were LR-HPVs and two were HR-HPVs. HPV6, HPV11, and an undetermined type were detected in 10, 4, and 1 patient(s), respectively, among the LR-HPVs. HPV16 and HPV35 were detected in one patient each among the HR-HPVs. Two patients with LSP were concurrently infected with LR- and HR-HPVs. Among these, one patient demonstrated concurrent infection with HPV11 and HPV35, whereas another patient exhibited concurrent infection with HPV6, HPV11, and HPV16.

The HPV genotypes included LR and HR types in 4 patients and 9 patients, respectively, among the 13 patients with HPV infection in OPSP, indicating a 9.7% prevalence of HR-HPV infection in patients with OPSP. HPV16 was the most prevalent genotype (*n* = 7, 53.8%), followed by HPV11 (*n* = 2, 15.4%) and then, HPV6, HPV39, HPV58, HPV66, and HPV84 (*n* = 1, 7.7% for each). The HPV genotypes included LR and HR types in 14 patients and 2 patients, respectively, among the 14 patients with HPV infection in LSP, indicating a 7.4% prevalence of HR-HPV infection in patients with LSP. HPV6 was the most prevalent genotype (*n* = 10, 71.4%), followed by HPV11 (*n* = 4, 28.4%), HPV16, HPV35, and an undetermined type (*n* = 1, 7.1% for each). The prevalence of HR-HPV infection was not statistically different between the OPSP and LSP groups (*p* = 1.000) (Table 2). However, the proportion of HR types in all HPV infections was significantly higher in the OPSP group than in the LSP group (69.2% vs. 14.3%, *p* = 0.004).

P16 status was assessed in 48 patients with OPSPs or LSPs. For 27 SPs with HPV infection (13 HPV(+)-OPSPs and 14 HPV(+)-LSPs), p16 IHC was performed as a routine examination to evaluate the transcriptional activity of their HPV infection. For the remaining 21 SPs without HPV infection (19 HPV(−)-OPSP and 2 HPV(−)-SP), p16 IHC was performed as a negative control to confirm the negative results of p16 IHC. The pathological examination revealed negative p16 IHC in all SPs, regardless of HPV infection status and the sites (Table 2).

### 3.3. Prevalence of HPV Infection According to Clinicodemographic Characteristics

The prevalence of overall HPV infection in OPSPs was 12.2% and 10.0% in patients aged <51 and ≥51 years (*p* = 0.699), 9.5% and 14.3% in males and females (*p* = 0.522), 12.9% and 8.8% in never-smokers and ex- or current smokers (*p* = 0.470), and 10.7% and 12.5% in solitary and multiple tumors, respectively (*p* = 0.687). The prevalence of overall HPV infection in LSP was 10.2% and 12.9% in patients aged <51 and ≥51 years (*p* = 0.658), 14.3% and 5.7% in males and females (*p* = 0.228), 16.1% and 7.0% in never-smokers and ex- or current smokers (*p* = 0.239), 7.8% and 37.5% in solitary and multiple tumors (*p* = 0.004), and 0.9% and 85.7% in non-recurrent and recurrent tumors (*p* < 0.001), respectively (Table 3).

The prevalence of HPV infection was investigated in 26 patients with LSPs according to the presence of dysplasia on pathological examination. The prevalence of HPV infection was 67.7% (4/6) in LSPs with dysplasia and 10.0% (2/20) in LSPs without dysplasia (*p* = 0.013). HPV6 was the only genotype identified in these LSPs with HPV infection.

### 3.4. Prevalence of HPV Infection According to Anatomical Subsites

OPSP was distributed in the following oropharyngeal subsites: tonsil (*n* = 38), tongue base (*n* = 18), soft palate (*n* = 29), and posterior wall (*n* = 12), with a prevalence of overall HPV infection of 15.8%, 11.1%, 6.9%, and 25.0% (*p* = 0.418), HR-HPV infection of 13.1%, 11.1%, 3.4%, and 8.3% (*p* = 0.583), and HPV16 infection of 10.5%, 5.6%, 3.4%, and 8.3% (*p* = 0.732), respectively. The proportions of HR-HPV among all HPV infections at the same subsites were 83.3%, 100.0%, 50.5%, and 66.7%, respectively (*p* = 0.305) (Figure 2).

LSP was distributed in the following laryngeal subsites: supraglottis (*n* = 14) and glottis (*n* = 12), with the prevalence of overall HPV infection of 35.7% and 75.0%, indicating a statistically significant difference (*p* = 0.045), HR-HPV infection of 7.1% and 8.3% (*p* = 1.000), and HPV16 infection of 7.1% and 0.0% (*p* = 0.418), respectively. The proportions of HR-HPV among all HPV infections at the same subsites were 20.0% and 11.1%, respectively (*p* = 1.000) (Figure 2).

### 3.5. Clinicodemographic Characteristics of HPV(+)-OPSP and HPV(+)-LSP

The mean ages of patients with HPV(+)-OPSP and HPV(+)-LSP were 46.9 ± 17.4 and 54.3 ± 17.1 years, respectively (*p* = 0.277). The proportions of males and females were 61.5% (*n* = 8) and 38.5% (*n* = 5) in patients with HPV(+)-OPSP and 85.7% (*n* = 12) and 14.3% (*n* = 2) in those with HPV(+)-LSP, respectively (*p* = 0.209). Further, never- and ex- or current smokers were reported in eight (61.5%) and five (38.5%) patients with HPV(+)-OPSP and ten (71.4%) and four (28.6%) in those with HPV(+)-LSP (*p* = 0.695), respectively. Furthermore, 11 (84.6%) and 8 (57.1%) cases of HPV(+)-OPSP and HPV(+)-LSP presented as solitary tumors, and 2 (15.4%) and 6 (42.9%) cases of HPV(+)-OPSP and HPV(+)-LSP presented as multiple tumors (*p* = 0.209), respectively (Table 4).

## 4. Discussion

This study delineated the prevalence and characteristics of HPV infection in SPs across two distinct anatomical sites: the oropharynx and larynx. The oropharynx and larynx demonstrated a marked disparity in the prevalence and genotypes of HPV infection, indicating the substantial heterogeneity in HPV infection between these sites.

This study revealed a significant difference in the prevalence of overall HPV infection between OPSP and LSP (14.0% vs. 53.8%), with a 3.8-fold higher prevalence of HPV infection in LSP than in OPSP. This result is consistent with previous epidemiological studies, although only a few demonstrated comparative data at these sites [16,17,18,19]. Earlier studies, including a Japanese study, which was the only study that conducted a direct comparison between OPSP and LSP, reported a prevalence of 0% (0/18) for HPV infection in OPSP and 66.7% (10/15) in LSP [18]. Similarly, among studies that included the oropharynx and larynx as serial sites for head and neck SPs, an Italian study in 2017 revealed a prevalence of 6.9% (3/43) and 66.7% (4/6) for HPV infection in OPSP and LSP, respectively, with a 9.7-fold difference in prevalence [17]. Additionally, a United States study in 2020 revealed a prevalence of 6% (4/63) and 64% (30/47) for HPV infection in OPSP and LSP, respectively, with a 10.7-fold difference in prevalence [16]. Indeed, the current study revealed a relatively small degree of difference in prevalence compared with previous studies. This discrepancy was caused by the relatively high prevalence of HPV infection in OPSP in the present study compared to prior studies. This study, which includes the highest number of OPSP cases to date and ensures uniformity in epidemiological background and methodology, brings our results closer to the actual prevalence. However, the prevalence of LSP in this study was comparable with or slightly lower than those in previous studies, possibly because our study included all LSP types, in contrast to earlier studies that primarily included RRP, which is known to have a high HPV prevalence. In addition, the prevalence of HPV infection was significantly higher in LSPs with dysplasia, with HPV6 as the only genotype. These findings were similar to the study by Davids et al., which showed 62% of laryngeal dysplasia cases were positive for HPV6 or 11, and to the study by Hall et al., which found HPV6 to be the most common subtype in LSPs [20,21]. However, the exact role of HPV in the development of laryngeal dysplasia remains controversial due to conflicting results on reported HPV prevalence and lack of pathogenesis in terms of cause-and-effect relationship.

The difference in prevalence between OPSP and LSP may be attributed to the viral biology of HPV, which demonstrates epitheliotropic and highly site-specific features [22,23]. HPV, which is an intranuclear virus, infects basal cells in the actively mitotic epithelium, frequently at sites of mucosal damage, metaplastic epithelium, or squamous epithelium junctions [24,25]. The larynx, which has a squamociliary junction that comprises stratified squamous and pseudostratified ciliated columnar epithelium, may be susceptible to HPV infection because of cell immaturity in this transition zone [24]. HPV infects the stratified epithelium, where basal cells are exposed through microepithelial injury, and cells incorporating episomal HPV may undergo multilayered proliferation and develop into SPs [23,24,25,26]. The present study revealed a 53.8% prevalence of HPV in LSP, indicating that this positive result for HPV DNA only demonstrates the presence of HPV, does not necessarily imply its role in tumorigenesis, and reflects the presence of nonviral mechanisms. However, the significantly higher prevalence of HPV in LSP than in OPSP emphasizes the influence of anatomical properties on the likelihood of HPV infection.

In this study, no HPV(+)-SPs showed positivity for p16 IHC, even in patients with HR-HPV infections. Discordance between HPV and p16 is not a rare finding, even in OPSCCs in which up to 30~40% showed p16(+)/HPV(−) or HPV(+)/p16(−) results [27,28]. P16 overexpression is an excellent surrogate marker of the biologically active HPV infection because functional inactivation of the retinoblastoma protein (Rb) by HPV E7 protein leads to p16 overexpression [8,9,13]. However, it has been well documented that all HPV infections cannot lead to transcriptionally active infections but can result in heterogeneous phenotypes depending on the association with subsequent genetic and signaling alterations [29,30]. HPV infection is usually self-limiting without any HPV-induced tumorigenesis in most patients, may be associated with HPV-induced benign tumors in some patients, or may drive HPV-induced carcinogenesis in very few patients [13,30]. Therefore, the current findings that none of the HPV(+)-SPs showed positivity for p16 IHC, even in patients with HR-HPV infections, indicate that none of the HPV infections in this study had transcriptional activity, which could drive HPV-induced carcinogenesis.

This study detected the genotypes in OPSP of HPV6, HPV11, HPV84, HPV16, HPV39, HPV58, and HPV66, with HPV16 being the most prevalent (53.8%, 7/13), indicating the predominance of the HR groups (69.2%, 9/13). In contrast, the genotypes detected in LSP were HPV6, HPV11, undetermined LR, HPV16, and HPV35, with HPV6 being the most prevalent (71.4%, 10/14), demonstrating the predominance of LR groups (100.0%, 14/14). These results differ from previous studies, which revealed that LR genotypes, including HPV6, were most frequently detected in all head and neck SPs, regardless of the anatomical site [16,31]. A study on 129 head and neck SPs, including 59 OPSPs and 17 LSPs, reported HPV6 (71%) and HPV11 (26%) as the most frequent genotypes, with no difference across the anatomical sites [16]. A previous study revealed that the detection of HR genotypes in LSP was co-infected with HPV6 or HPV11, with HPV16 and 35 each identified in one case [10]. However, the literature on the detection of HR genotypes in OPSPs is lacking, and only one previous study has revealed the detection of HPV16 in OPSPs [13]. Nevertheless, earlier studies on HPV infection in the oropharynx of unvaccinated, tumor-free healthy patients revealed HPV16 as the most commonly detected genotype. Furthermore, HPV16 was the most responsible in HPV-positive OPSCC, accounting for 85–96% of cases [32,33,34,35]. Thus, these results reveal that the oropharynx is a critical site for HPV16 infection, regardless of whether the pathological phenotype resulting from viral infection is tumor-free, benign, or malignant.

The distribution of HR-HPVs in our study, particularly HPV 16, is relevant in the context of HPV vaccination as a preventive strategy against HPV-related cancer. The proportion of HR-HPV was significantly higher in OPSP (69.2%) than in LSP (14.3%), consistent with the trend of HPV contribution in head and neck squamous cell carcinoma (HNSCC) across different cancer sites [36,37]. A large 2017 international study on 3,680 HNCs revealed that the fraction of attribution of HPV was estimated as 22.4% in oropharyngeal cancer, demonstrating a significant contribution of HPVs, whereas it was estimated as 3.5% in laryngeal cancer [37]. Another study in 2023 revealed that 60% of 240 OPSCCs and only 10% of non-OP HNSCCs were p16-positive, indicating no significance in the involvement of HPV in non-OP HNSCC than that in OPSCC [36]. The association between the HR-HPV distribution in SPs and HPV-related HNC is unknown. Based on the natural history of oral HPV infection, 70% of HPV infections resolve within 2 years, whereas HPV16 infection lasts for 5 years [38,39]. Therefore, this distribution of HR-HPVs, including HPV16, in OPSP and LSP indicates a high likelihood of persistent infection, which is the initiation of HPV-driven carcinogenesis. The results of this study, particularly in the era of HPV vaccination, could predict the actual contribution of HPV to HNSCC.

The present study revealed no significant differences in the prevalence of HPV infection between patients with OPSP and LSP according to clinicodemographic characteristics, including age, sex, and smoking status. To date, very few studies have analyzed the association between the prevalence of HPV infection and clinicodemographic factors in SPs [13,18]. However, HPV infection is more prevalent in males, older individuals, and those who currently smoke. This is according to the current consensus regarding HPV infection in the head and neck mucosa, including the oral cavity and oropharynx [40,41,42]. The prevalence of HPV infection was significantly higher in those with multiple or recurrent tumors in patients with LSP. This result is consistent with the findings of previous studies indicating that RRP, defined as contiguous papilloma with a pattern of multiple recurrences among LSP, is a viral disease caused by HPV [10,15,43].

Despite being the most extensive study focusing on OPSP and LSP, the present study has some limitations. The first is the lack of representativeness of the included samples of OPSP and LSP. Only OPSP and LSP with HPV status were included, considering that HPV testing in patients with SP is not a routine diagnostic tool. Therefore, the small number of included samples limits the validity of our results. Furthermore, this study is not population-based. Thus, it does not represent the actual prevalence of HPV infection in OPSP and LSP in the general population. Second, the absence of comprehensive demographic, sexual, and sociocultural data restricts our understanding of HPV transmission dynamics because of this study’s retrospective nature. Thus, prospective population-based studies are required to provide a more accurate representation of HPV prevalence in these sites, and those data can be used for establishing and evaluating national health policies, such as the cost-effectiveness of HPV vaccination strategies.

## 5. Conclusions

In summary, this study revealed the highly heterogeneous prevalence and characteristics of HPV infection in OPSP and LSP. The prevalence of overall HPV infection was 14.0% and 53.8% in OPSP and LSP, respectively, and the most prevalent genotype was HPV16 and HPV6, respectively. The proportions of HR in all HPV infections were 69.2% and 14.3% in OPSP and LSP, respectively; thus, the burden of HR-HPV was relatively higher in OPSP, consistent with prevalence of HPV-related HNSCC. The results of our study could improve understanding of HPV infection in SP of the oropharynx and larynx and serve as valuable epidemiological data for HPV-related tumorigenesis of the oropharynx and larynx.

## Figures and Tables

**Figure 1 diagnostics-14-01163-f001:**
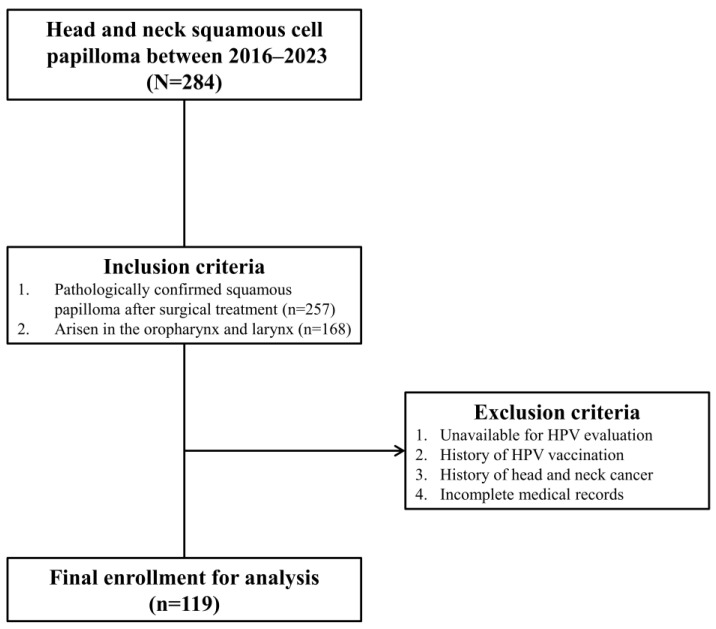
Patient enrollment flow chart.

**Figure 2 diagnostics-14-01163-f002:**
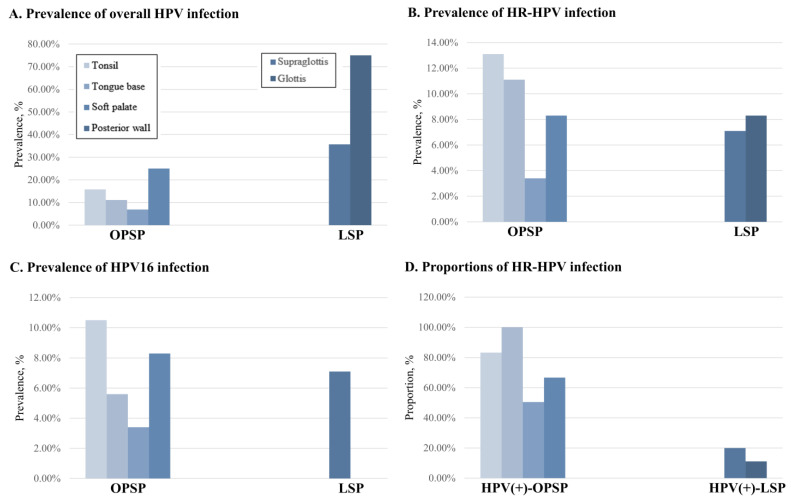
The oropharyngeal subsites included the tonsil, tongue base, soft palate, and posterior wall, and the laryngeal subsites were supraglottis and glottis. The prevalence of overall human papillomavirus (HPV) infection (**A**), high-risk (HR)-HPV infection (**B**), and HPV16 infection (**C**) according to oropharyngeal and laryngeal subsites. The proportions of HR-HPV infection among all HPV infections according to oropharyngeal and laryngeal subsites (**D**).

**Table 1 diagnostics-14-01163-t001:** Clinicodemographic characteristics.

Variables	OPSP (*n* = 93)	LSP (*n* = 26)	*p*-Value
Age (years)	50.3 ± 16.5	55.3 ± 15.5	0.171
Sex			
Men	61 (65.6%)	23 (88.5%)	0.024
Women	32 (34.4%)	3 (11.5%)	
Smoking status			-
Never	49 (52.7%)	13 (50.0%)	0.867
Ex	13 (14.0%)	5 (19.2%)	
Current	31 (33.3%)	8 (30.8%)	
Clinical presentation			
Symptomatic	29 (31.2%)	16 (61.5%)	0.005 ^a^
Voice change	0 (0.0%)	14 (53.9%)	
Sore throat	5 (5.4%)	0 (0.0%)	
Foreign body sensation	22 (23.7%)	1 (3.8%)	
Chronic chough	2 (2.1%)	1 (3.8%)	
Asymptomatic	64 (68.8%)	10 (38.5%)	
By self-examination	14 (15.0%)	0 (0.0%)	
By dental examination	1 (1.1%)	0 (0.0%)	
By endoscopic examination	49 (52.7%)	10 (38.5%)	
Focality			
Solitary	87 (93.5%)	16 (61.5%)	<0.001
Multiple	6 (6.5%)	10 (38.5%)	
Dysplasia on pathologic examination			
No	93 (100.0%)	20 (76.9%)	<0.001
Yes	0 (0.0%)	6 (23.1%)	
Recurrence			
No	93 (100.0%)	19 (73.1%)	<0.001
Yes	0 (0.0%)	7 (26.9%)	

OPSP, oropharyngeal squamous cell papilloma; LSP, laryngeal squamous cell papilloma. ^a^ *p*-value to indicate comparison between symptomatic and asymptomatic tables may have a footer.

**Table 2 diagnostics-14-01163-t002:** Prevalence of HPV infection and genotypes.

Variables	OPSP (*n* = 93)	LSP (*n* = 26)	*p*-Value
Overall HPV infection			
Negative	80 (86.0%)	12 (46.2%)	<0.001
Positive	13 (14.0%)	14 (53.8%)	
Genotypes			
Low-risk	4 (4.3%)	14 (53.8%)	<0.001
6	1 (1.1%)	10 (1.4%) ^c^	
11	2 (2.2%)	4 (15.4%) ^b,c^	
84	1 (1.1%)	0 (0.0%)	
Undetermined	0 (0.0%)	1 (3.8%)	
High-risk	9 (9.7%)	2 (7.4%)	1.000
16	7 (7.5%)	1 (3.8%) ^c^	
35	0 (0.0%)	1 (3.8%) ^b^	
39	1 (1.1%) ^a^	0 (0.0%)	
58	1 (1.1%)	0 (0.0%)	
66	1 (1.1%) ^a^	0 (0.0%)	
Immunohistochemistry for p16 (*n* = 48) ^d^			
Negative	32 (100.0%)	16 (100.0%)	-
Positive	0 (0.0%)	0 (0.0%)	

OPSP, oropharyngeal squamous cell papilloma; LSP, laryngeal squamous cell papilloma; HPV, human papillomavirus. ^a^ one patient with OPSP coinfected with HPV39 and HPV66. ^b^ one patient with LSP coinfected with HPV11 and HPV35. ^c^ one patient with LSP coinfected with HPV6, HPV11, and HPV16. ^d^ P16 status was assessed in total 48 patients with OPSPs or LSPs, including 13 HPV(+)-OPSPs and 14 HPV(+)-LSP. Multiple concurrent infections were counted as double or triple, which explains why the sum is higher than the overall HPV infection rate.

**Table 3 diagnostics-14-01163-t003:** Prevalence of HPV infection in OPSP and LSP according to clinicodemographic characteristics.

	Age			Sex			Smoking			Focality			Recurrence		
	<51 (*n* = 49)	≥51 (*n* = 70)	*p*-Value	Men (*n* = 84)	Women (*n* = 35)	*p*-Value	Never (*n* = 62)	Ex- or Current (*n* = 57)	*p*-Value	Solitary (*n* = 103)	Multiple (*n* = 16)	*p*-Value	Non-Recurrent (*n* = 112)	Recurrent (*n* = 7)	*p*-Value
Overall HPV(+) in OPSP	12.2% (6/49)	10.0% (7/70)	0.699	9.5% (8/84)	14.3% (5/35)	0.522	12.9% (8/62)	8.8% (5/57)	0.470	10.7% (11/103)	12.5% (2/16)	0.687	-	-	-
Overall HPV(+) in LSP	10.2% (5/49)	12.9% (9/70)	0.658	14.3% (12/84)	5.7% (2/35)	0.228	16.1% (10/62))	7.0% (4/57)	0.239	7.8% (8/103)	37.5% (6/16)	0.004	0.9% (1/112)	85.7% (6/7)	<0.001

OPSP, oropharyngeal squamous cell papilloma; LSP, laryngeal squamous cell papilloma; HPV, human papillomavirus.

**Table 4 diagnostics-14-01163-t004:** Clinicodemographic characteristics of HPV(+)-OPSP and HPV(+)-LSP.

Variables	HPV(+) OPSP (*n* = 13)	HPV(+) LSP (*n* = 14)	*p*-Value
Age	46.9 ± 17.4	54.3 ± 17.1	0.277
Sex			
Men	8 (61.5%)	12 (85.7%)	0.209
Women	5 (38.5%)	2 (14.3%)	
Smoking			
Never	8 (61.5%)	10 (71.4%)	0.695
Ex- or Current	5 (38.5%)	4 (28.6%)	
Focality			
Solitary	11 (84.6%)	8 (57.1%)	0.209
Multiple	2 (15.4%)	6 (42.9%)	
Genotype distribution			
LR	4 (30.8%)	14 (100.0%)	<0.001
HR	9 (69.2%)	2 (14.3%)	0.004

OPSP, oropharyngeal squamous cell papilloma; LSP, laryngeal squamous cell papilloma; HPV, human papillomavirus; HR, high-risk; LR, low-risk; 9-valent, 9-valent vaccine types.

## Data Availability

The data that support the findings of this study are available from the corresponding author on reasonable request.

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
