# Peer review of "Human Papillomavirus Infection of the Oropharyngeal and Laryngeal Squamous Papilloma: Disparities in Prevalence and Characteristics"

_diagnostics, 2024, doi:10.3390/diagnostics14111163_

Round 1

Reviewer 1 Report (Previous Reviewer 1)

Comments and Suggestions for Authors

The submitted manuscript describes the infection rate of HPVs and the proportions of genotypes in oropharyngeal squamous papilloma (OPSP) and laryngeal squamous papilloma (LSP).

The manuscript has been well revised. The reviewer thinks the present manuscript is acceptable.

Reviewer 2 Report (Previous Reviewer 3)

Comments and Suggestions for Authors I have reviewed the revised manuscript. It is much improved and the criticisms have been addressed. I would suggest to proceed with accepting for publication after the usual spelling and grammar checks

Comments on the Quality of English Language I would suggest to proceed with accepting for publication after the usual spelling and grammar checks

This manuscript is a resubmission of an earlier submission. The following is a list of the peer review reports and author responses from that submission.

Round 1

Reviewer 1 Report

Comments and Suggestions for Authors

The submitted manuscript describes the infection rate of HPVs and the proportions of genotypes in oropharyngeal squamous papilloma (OPSP) and laryngeal squamous papilloma (LSP).

Although the reviewer considered the topic interesting and falls within the scope of diagnostics, the reviewer recommends considering additional discussion on the following points.

1.       The author should make a reasonable interpretation for no immunoreactivity in immunohistochemistry for p16. Additionaly, detailed information about evaluated cases for p16 immunohistochemistry should be provided in table 2.

2.       The reviewer recommends adding histological presentations of OPSP and LSP with a discussion of characteristic histological findings of HPV infection, e.g., koilocytotic atypia.

3.       The reviewer recommends discussing the association of the pathogenesis of dysplasia and HPV infection status in LSPs.

Author Response

please find the response to reviewer #1 in the attachment.

Reviewer 2 Report

Comments and Suggestions for Authors

The manuscript clear and well-structured, correctly written from a scientific point of view. The published data is presented correctly and coherently, accompanied by tables that are easy to read and understand, properly analyzed scientifically. The number of patients is small and allows only conclusions with relative epidemiological value. References are mostly recent or very recent and I did not notice elements of plagiarism. The manuscript does not highlight new elements, but only purely epidemiological data, already known

Author Response

please find the response to reviewer #2 in the attachment.

Reviewer 3 Report

Comments and Suggestions for Authors

Dear Authors, thanks for your contribution.  These are my comments

Fig 2 can be omitted - the same information is presented in Table 2

Please explain how many specimens were stained for p16?  In Table 2, it states 32 Negative for p16 in 93 OPSP specimens?  Why were the other specimens not stained for p16?  Same for LSP 16 p16 negative of 26  specimens - what about the other 10 LSP?

Please include in your discussion why specimens positive for HR HPV but p16 negative?  Because in HPV positive OPC, we see p16 stain positive.

In your results, line 141, it states 14 LSP patients were positive for HPV but in Table 4 under HPV+ LSP N=13 - please explain the discrepancy.

The discussion is too long.  Please edit and shorten this by 30 - 50%.  In particular the paragraph HPV vaccination (lines 286 - 301) does not contribute to the manuscript and could be omitted.  

Comments on the Quality of English Language

There are a moderate number of English grammar errors that require proofreading and edits.

Author Response

please find the response to reviewer #3 in the attachment.
